# Effects of Functional Strength Training on Functional Movement and Balance in Middle-Aged Adults

**Ozkan Guler** [1,*] , **Oguzhan Tuncel** [2] **and Antonino Bianco** [3]

1   Department of Coaching Education, Faculty of Sports Science, Ankara University, Ankara 06560, Turkey
2   Department of Coaching Education, School of Physical Education and Sports, Iğdır University, Iğdır 76000, Turkey; oguzhan.tuncel@igdir.edu.tr
3   Department of Psychology, Educational Science and Human Movement, Sport and Exercise Sciences Research Unit, University of Palermo, 90144 Palermo, Italy; antonino.bianco@unipa.it
*   Correspondence: oguler@ankara.edu.tr

**Abstract:** Functional movement deficiencies cause falls and injuries in adults. Functional strength training (FST) is emerging as a new training method for athletes, middle-aged and older adults, to improve functional movement: The present study was conducted in order to investigate the effects of FST on balance and functional movement in healthy and independent middle-aged adults. The sample for this study consisted of 46 physically active individuals (24 female and 22 male). A total of 46 subjects were divided based on randomly into the functional strength training (FST) group ($n = 26$) aged: 51.55 ± 3.73 years; height: 168.69 ± 8.8 cm; body mass: 75.88 ± 12.18; and traditional strength training (TST) group ($n = 20$) age: 52.85 ± 4.01; height: 166.9 ± 9.98; body mass: 76.15 ± 10.45. Each group performed 24 sessions of a training protocol three-time a week. The functional movement was assessed using the functional movement screen (FMS) protocol. Balance performance was determined by using the balance error scoring system (BESS). Bodyweight and body fat ratio were measured using bioelectric impedance. There was a significant statistical difference between FMS total scores after an eight-week FST in the FST group. After the intervention, the functional strength group tended to have significantly better balance control than the traditional strength group ($p = 0.01$). Statistically, significant differences were observed between pre-test and post-test in the intervention group on BMI, body fat, and body mass ($p = 0.01$). There were not found significant differences in balance control and FMS score in TST group. As a result of this study, FST positively affected the FMS total score and balance performance in middle-aged adults. Early detections of the deficiencies in functional movement and balance in the middle ages may reduce the risk of insufficiency and fall in adults through targeted functional strength training intervention.

**Keywords:** adults; balance; fall risk; FMS; functional strength

## 1. Introduction

Aging is associated with decreased overall muscle strength caused by many factors, such as neural changes and sarcopenia. Muscle losses usually start at the age of 30 and increase, especially after 60 years [1]. Long term studies have shown that muscle mass decrease varies between 1 and 1.4 per year in the lower limbs, and this rate is higher than the losses in the muscle of the upper limb [2–4]. The decrease in muscle strength with age is 2–5 times higher than muscle mass loss [5]. Decreases in muscle strength and mass in middle-aged adults may negatively affect the quality of daily activity applications (walking, sitting, climbing stairs) that performed in the advanced ages [6–8]. According to meta-analysis results, strength training is seen as an effective way to eliminate strength loss in middle-aged adults [9]. There is strong evidence that strength training reduces the effects of aging on the neuromuscular system and functional capacity [10–15]. Strength training elicits the reduction of adipose tissue [16] and increases muscle mass [17,18], bone density [19], and muscle strength [20,21]. Traditional strength training provides an increase

in muscle strength. However, it does not necessarily cause significant changes in the development of functionality and the performance of daily tasks [22,23].

For this reason, novel strength training methods are needed to improve functional movement in adults. Functional strength training (FST) is emerging as a new training method for athletes, middle-aged adults, elderly, and cardiac patients [24,25]. It is also known to positively affect the performance of daily activities, health, and weight control [26,27]. FST works by simulating targeted movements that require strength, flexibility, balance, and coordination [28,29].

Functional strength training aims to move in multiple planes and develop multiple muscle groups with a single exercise. FST involves agility drills, closed-chain and multi-directional exercises, ballistic movements such as medicine ball throwing, and balance activities that target physiological systems and neuromuscular systems [30]. In FST, movements based on simulating daily life activities are performed [31]. As a result of FST, the targeted movement develops rather than specific muscle development [32]. In order to perform FST in a more effective and target-oriented manner, deficiencies in the functional movement must be correctly identified. Functional movement screen (FMS) is the most common test for evaluating functional movement deficiencies for individuals. The FMS consists of seven fundamental movement patterns that include balance, mobility, stability, and motor control [33]. These fundamental movement patterns are designed to observe basic locomotor, manipulative, and movement stabilization [34]. FMS can be used efficiently for determining loss of strength, instability, and immobility

Detecting functional movement deficiencies in the middle-aged group and performing FST can be helpful to minimize the loss of functionality in advanced ages. Previous studies are mostly focused on groups over the age of 65, but there are no adequate studies among middle-aged adults. Therefore, the present study aimed to investigate the effects of FST on balance control and functional movement in healthy and independent middle-aged adults.

## 2. Materials and Methods

A total of 46 physically active individuals who regularly workout at least 3 -time in a week for 6 months (24 females and 22 males) participated in this study. Participants were assigned randomly into the functional strength training (FST) group, n = 26 (aged: 51.55 ± 3.73 years; height: 168.69 ± 8.8 cm; body mass: 75.88 ± 12.18) and traditional strength training (TST) group n = 20 (age: 52.85 ± 4.01; height: 166.9 ± 9.98; body mass: 76.15 ± 10.45). At the beginning of the study and after eight-week of intervention, the participants were assessed according to their functional movement, balance control, and anthropometry. The tests included measurements of functional movement screen, balance, and body composition. The functional movement was assessed using the functional movement screen (FMS) protocol [33]. Balance performance was assessed by using the Balance error scoring system (BESS) [35]. Bodyweight and body fat ratio were measured using bioelectric impedance PlusAvis 333 (Jawon Medical, Seoul, Korea). Body mass index was calculated by dividing the body weight by height squared. Height was determined through a clinical stadiometer (Holtain Stadiometer, Holtain Ltd. Dyfed, UK). Eligibility criteria for participants were free from orthopedic dysfunction. The study was conducted according to the Declaration of Helsinki as reflected in a priori approval by the institution's human research committee (Approval code 20-1312-17).

Functional Movement:

FMS includes seven motor tasks: overhead deep squat, hurdle step, in-line lunge, shoulder mobility, active straight leg raise, rotatory stability test, and trunk stability push-up. The FMS using the standard 0–3 ordinal system that was fully described by Cook et al. [33]. A score of 3 indicates that specific movements were performed correctly without pain and compensation, a score of 2 was given if the movement was performed with compensatory movements observed, a 1 score was indicated subject could not complete movement, and a score 0 was given if any pain being present during the movement [34]. Each task was performed three times, and the best score was recorded for

further analysis. The FMS test was conducted by using standard equipment (FMS Test Kit, Functional Movement Systems Inc., Chatham, VA, USA).

Balance:

The balance error scoring system (BESS) consists of 3 stances: double-leg stance, single-leg stance, and a tandem stance in a heel-to-toe fashion. The stances are performed on a firm surface and a foam surface with the eyes closed and hands-on-hips. The number of errors counted during each 20-s trial [35]. Errors were counted for any of the following occurrences; opening eyes, lifting hands off hips changing their foot placement, hip flexion or abduction more than 30°, stepping, stumbling, falling, or return the test position for longer than 5 s. A higher total test score in the BESS indicates poor balance performance.

Strength Training:

Both groups performed 24 training sessions (three days a week for eight weeks). All participants performed the training program at the commercial fitness center. Exercise intensity in both the functional strength and traditional strength program was set a 6–7 on a 10-point rated perceived exertion (RPE) scale [36,37]. To maintain RPE of 6–7 at each session, intensity or volume was gradually lifted by individual performance.

Functional strength training consists of three blocks for each training session. These blocks included stability, strength, and intermittent exercises. Examples of some functional strength exercises are presented in Figure 1. Each training session lasted approximately 60 min. Warm-up section for 10 min and dynamic mobility exercises for major joints. 10 min of stability exercises, 15 min of multiple joint strength exercise for strength development, 15 min of intermittent activities (agility and coordination sprint interval and rope pulling), 10 min cooldown period of flexibility exercise for limbs and trunk.

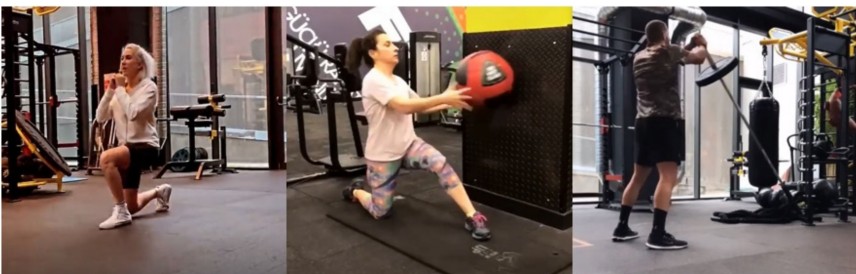

**Figure 1.** Example of functional strength exercise.

Traditional strength training was divided into three blocks. These blocks consist of upper body, lower body and trunk exercises. Full body strength training method was used for the traditional strength group. Participants performed resistance exercises machine-based with isolated neuromuscular work. The following traditional strength exercises were performed: Bench press, seated row, lat pull down, biceps curl and triceps pushdown, leg extension, leg flexion, leg press, sit up, and back hyperextension. Examples of some traditional strength exercises are presented in Figure 2. Each workout is divided into segments: 10 min for warm-up and 40 min for main TST protocol and 10 min for stretching and cool down. Each exercise was repeated in 3 sets, 8–12 repetitions, and max 120-s rest were given between sets.

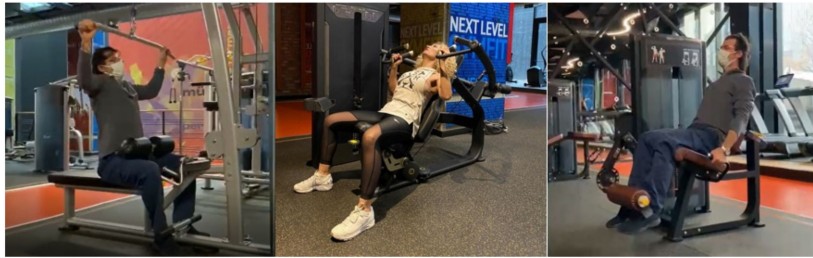

**Figure 2.** Example of traditional strength exercise.

Statistical Analysis:

The data of the intervention and control groups were expressed with descriptive statistics with mean and standard deviation. The normality distribution of the data was determined by the Shapiro–Wilk normality statistical method. It showed a normal distribution for both groups on both tests. An independent sample t-test was used to determine differences between the intervention and control groups. Paired sample *t*-tests were utilized to determine pre-post differences within each group. The significant Alpha levels for the whole procedure was set at $p < 0.05$. Data were analyzed using the Statistical Package for Social Sciences (SPSS) version 23.

## 3. Results

The baseline characteristics of the participants are shown in Table 1. The mean age was $51.55 \pm 3.73$ in the FST group and $52.85 \pm 4.01$ in the TST group. There was no significant difference between the groups in baseline demographic variables.

**Table 1.** Baseline anthropometric characteristics of study participants.

| | FST (n = 26) | TST (n = 20) | Total Participants (n = 46) |
|---|---|---|---|
| Age (years) | $51.55 \pm 3.73$ | $52.85 \pm 4.01$ | $52.08 \pm 3.89$ |
| Body height (cm) | $168.69 \pm 8.8$ | $166.9 \pm 9.98$ | $167.9 \pm 9.27$ |
| Body mass (kg) | $75.88 \pm 12.18$ | $76.15 \pm 10.45$ | $76.01 \pm 11.34$ |
| Body mass index (kg $\times$ m$^{-2}$) | $25.52 \pm 2.56$ | $25.89 \pm 6.20$ | $26.82 \pm 2.08$ |
| Body Fat% | $27.61 \pm 3.29$ | $26.6 \pm 3.6$ | $27.15 \pm 3.43$ |

Data are presented as mean $\pm$ SD; FST: Functional Strength Training; TST Traditional Strength Training.

The FST group showed significant improvement in BMI, body fat and body mass compared with the TST group ($p < 0.001$). In the TST group, no significant change was observed in body composition parameters through the intervention period. FMS total score and BESS between FST group and TST group are presented in Table 2. When the pre-test and post-test values of the group were considered, there were significant statistical differences between FMS total scores after an eight-week intervention in the FST group ($p = 0.001$) In contrast, the TST group showed no improvement in functional outcomes ($p > 0.05$). In the FST group, there was a significant improvement in the BESS result ($p = 0.001$). In the TST group, no significant difference was observed in BESS results ($p > 0.05$).

**Table 2.** Changes in outcomes of FMS and BESS 8-week intervention period.

| | | Within-Group Differences | | | | | | | Between-Group Differences | | |
|---|---|---|---|---|---|---|---|---|---|---|---|
| | FST (n = 26) | | | | TST (n = 20) | | | | TST | FST (n = 20) | |
| | Pre (Mean SD) | Post (Mean SD) | % | p | Pre (Mean SD) | Post (Mean SD) | % | p | Difference | Difference | p |
| FMS | $12.65 \pm 1.5$ | $15.11 \pm 1.72$ | 19.45 | 0.001 | $12.7 \pm 1.75$ | $12.95 \pm 1.97$ | 1.97 | 0.204 | $-2.5 \pm 1.72$ | $0.25 \pm 0.85$ | 0.001 |
| BESS | $14.92 \pm 2.91$ | $12.46 \pm 2.31$ | $-16.49$ | 0.001 | $14.45 \pm 2.41$ | $14.1 \pm 1.77$ | $-2.42$ | 0.130 | $2.46 \pm 1.98$ | $0.35 \pm 0.98$ | 0.001 |
| Body mass | $75.68 \pm 12.18$ | $73.53 \pm 11.95$ | $-2.84$ | 0.001 | $76.15 \pm 10.45$ | $76.66 \pm 10.17$ | 0.69 | 0.047 | $2.34 \pm 2.27$ | $0.5 \pm 1.05$ | 0.001 |
| BMI | $26.52 \pm 2.56$ | $25.69 \pm 2.55$ | $-3.12$ | 0.001 | $25.89 \pm 6.2$ | $26.09 \pm 6.25$ | 0.77 | 0.053 | $0.82 \pm 0.79$ | $0.25 \pm 0.63$ | 0.001 |
| Body Fat% | $27.61 \pm 3.29$ | $23.66 \pm 3.45$ | $-14.3$ | 0.001 | $26.6 \pm 3.6$ | $26.27 \pm 3.24$ | $-1.24$ | 0.32 | $3.95 \pm 4.1$ | $0.32 \pm 1.42$ | 0.001 |

Data are presented as mean $\pm$ SD; SD: Standard Deviation; FMS: Functional movement screen; BESS: Balance error scoring system; BMI: Body Mass Index.

FMS total score group differences between FST and TST were presented in Figure 3. No significant differences were observed for FMS measures between the FST and TST groups at baseline. When comparing the FST group and TST group, there was a significant improvement in FMS score in FST group ($p = 0.001$). BESS value group differences between FST and TST are presented in Figure 4. No significant differences were observed for BESS measures between the FMS and TST groups at baseline. BESS score showed significant

differences between FST group and TST ($p$ = 0.001). In addition, there was a significant difference in body composition parameters between FST group and TST group ($p$ < 0.05).

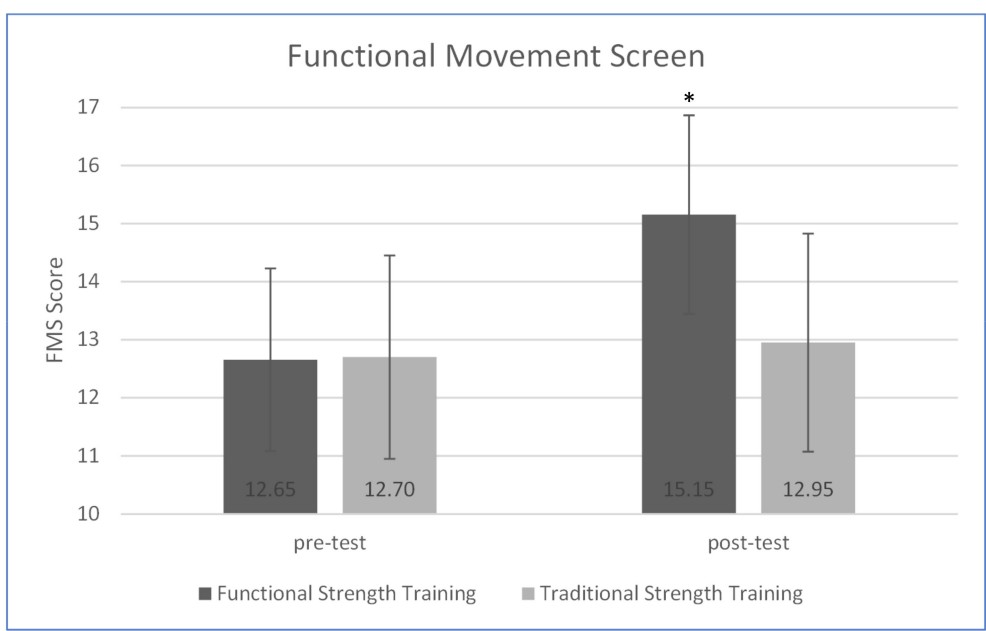

**Figure 3.** Comparison of Functional Movement Screen(FMS) score of Traditional Strength Training (TST) group and Functional Strength Training (FST) group. All values are presented as mean ± SE. * represent the significant difference (independent sample t-test) between FST group and TST group in post-test of FMS score ($p$ = 0.001).

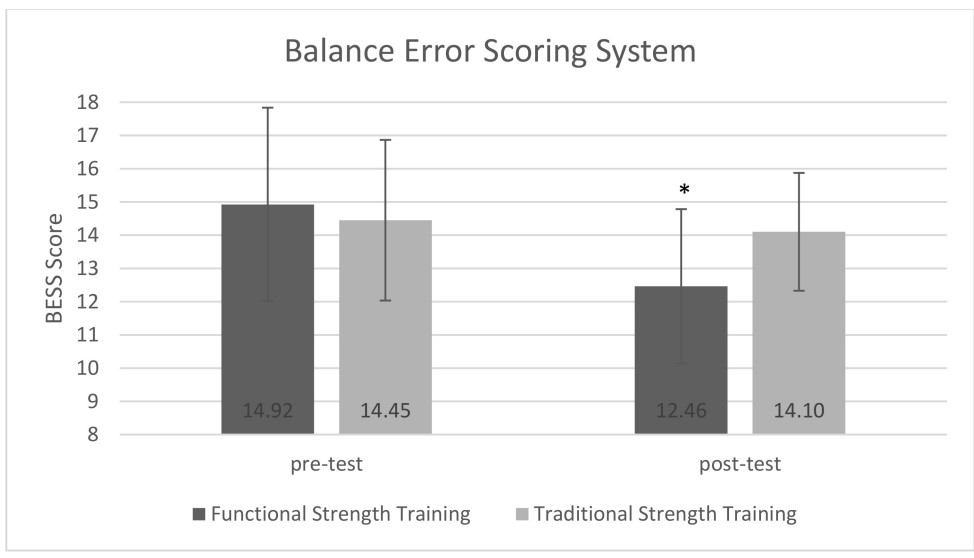

**Figure 4.** Comparison of Balance Error Scoring System (BESS) score of Traditional Strength Training (TST) group and Functional Strength Training (FST) group. All values are presented as mean ± SE. * represent the significant difference (independent sample *t*-test) between FST group and TST group in post-test of FMS score ($p$ = 0.001).

## 4. Discussion

This study aimed to examine the effects of functional strength training in FMS scores and balance ability in middle-aged adults. The most noticeable result of this study was the statistically significant difference in FMS total score, and BESS score after eight weeks of functional strength training. Additionally, an improvement was observed in body

composition in the FST group. However, no statistically significant improvement was observed in FMS score, balance ability and body composition in the TST group.

Aging is a natural process of the body and is characterized by progressive muscle loss, decreased balance, gait, and functionality [38]. Strength training has a crucial role in reducing these losses with the effects of strength training on the adaptation of neuromuscular control and proprioception. In the present study, it was determined that eight weeks of functional strength training increased FMS score and balance control in middle-aged adults. These results are consistent with previous studies [39,40]. In another study conducted with young adults, FST directly affects the FMS total scores [41]. According to a meta-analysis review, a functional movement score of <14 is associated with the risk of injury [42,43]. In addition, poor FMS score indicates low range of motion, limited joint mobility, and movement control [33,44]. In the current study, after FST intervention, the average FMS total score increased from 12.65 to 15.11 in FST group. However, these increases were from 12.7 to 12.97 in TST group. This development can be explained by the fact that the multidirectional exercise in functional strength training requires stability, mobility strength, and coordination. On the contrary, traditional strength training has a limited effect on functional movement. As traditional strength training methods are performed predominantly unidirectionally and on the sagittal axis, only the relevant muscle develops [31]. Therefore, FST should be preferred in order to improve functional movement.

As a natural aging process, body mass, BMI, and body fat ratio increase, and lean body mass decreases [20,31]. Several studies suggest that strength training leads to a reduction in adipose tissue, an increase in muscle mass, bone mineral density, and muscle strength [16,17,19]. Despite the lack of changes in body composition in TST group, a significant change was observed in body mass ($p = 0.001$), and body fat ratio ($p = 0.001$) in FST group in the present study. The effects of functional strength training can justify these results. It is known that FST produces more muscular activation than traditional training. Increased muscle contraction during functional movement can lead to more energy consumption, and it may help to reduce fat mass. Some authors have also shown that multi-directional strength training provides structural changes in body mass and adipose tissue in adults [11,45]. Resende-Neto, A.G., et al. (2019) compared the effects of 8-week functional strength and traditional strength training in body composition in older women, noted that functional strength training reduced fat mass more effectively than traditional strength training [46]. In a study comparing multi-joint exercise and single-joint exercise, it was reported that multidirectional-joint exercise is more effective than single-joint exercises [20]. Contrary to the results of this study, there are studies in the literature reporting that functional strength training and traditional strength training have similar effects on body composition. It was found that traditional strength and functional strength training had a similar effect on body composition [31]

Previous studies indicate that age-related decreases in muscle strength affect motor control and balance performance [47–50]. Impaired balance control can prospectively predict future falls and injuries [51,52]. Therefore, early detection of changes in poor balance control can be useful in identifying protective strategies and reducing fall risk. In the present study, balance performance improved after eight weeks of FST intervention and FST group showed increases of 16.49% in balance control after the intervention. In turn, these increases were 2.42% for TST group. The findings of this study are consistent with the previous studies in the literature. In a study investigating the effects of functional versus traditional strength training, it was reported that balance score had improved more in the FST group than in the traditional strength group [53]. Similarly, it was reported that 12 weeks of FST improved lower body strength, balance, and coordination [54]. Balance is affected by neuromuscular and proprioceptive processes. Structural and functional deterioration of the neuromuscular system and physio motor abilities increase with aging. It can be concluded that improvement in balance performance after and FST intervention may result from exercise-induced neuromuscular and proprioceptive adaptations. The main limitation of this study is that the participants' maximal strength and flexibility were

not measured. Also, the study design did not include the investigation of the physiological mechanisms of FST. Another limitation is that the participants were not followed up with the diet program during the study.

## 5. Conclusions

As a result of this study, FST positively affected the FMS total score and balance performance in middle-aged adults. FST contributes to the development of individuals' mobility, stability, balance, and strength. The early detection of deficiencies in functional movement and balance in middle-aged adults may help reduce the risk of disability and fall in older adults by targeted functional strength training intervention. As a result, functional strength training can be a key factor for a sustainable healthy life in middle-ages.

**Author Contributions:** Conceptualization, O.G. and A.B.; methodology, O.G. and A.B.; software, O.T.; validation, O.G., O.T. and A.B.; formal analysis, O.T.; investigation, O.G. and O.T.; resources, O.G.; data curation, O.G. and O.T.; writing—original draft preparation, O.G.; writing—review and editing, O.G. and A.B.; visualization, A.B.; supervision, A.B.; project administration, O.G. and A.B.; funding acquisition, O.T. All authors have read and agreed to the published version of the manuscript.

**Funding:** This research received no external funding.

**Institutional Review Board Statement:** The study was conducted according to the guidelines of the Declaration of Helsinki, and approved by the Human Research Ethics Committee of Health Science Institute (Approval code 20-1312-17).

**Informed Consent Statement:** Informed consent was obtained from all subjects involved in the study.

**Data Availability Statement:** The data presented in this study are available on request from the corresponding author. The data are not publicly available due to ethical reasons.

**Conflicts of Interest:** The authors declare no conflict of interest.

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
