# Peer review of "Effects of Functional Strength Training on Functional Movement and Balance in Middle-Aged Adults"

_sustainability, doi:10.3390/su13031074_

Round 1
Reviewer 1 Report
This study investigated the effects of functional strength training on functional movement and balance in the middle-aged adults. The results of the study could be an important source of health care for middle-aged adults. However, there are some revision, so please checklist as below.
- Do you have any photo of functional strength training? If so, please include it.
- Please describe information about the equipment used in this research. For example, you will need to include information such as model name, manufacturer, and country of manufacture. Like the information you put into the FMS tool.
- It would have been better if you included a control group in your study design. This part is a bit disappointing. Why didn't you include it in your research design?
- Please delete p<0.05 in the discussion section.
- In discussion section, middle-aged individuals?, Revise to middle-aged adults. Match the middle-aged adults.
- Please delete old adults at the end of the conclusion section. This study was conducted on the middle-aged adults.
- In Figure 1, Traditional StrengthS?, Please revise it.
- In Figure 1, please revise from Functional Strength to Functional Strength Training. If the word is long, revise it to FST.
- In Figure 1, please revise from Traditional Strength to Traditional Strength Training. If the word is long, revise it to TST.
- Please revise the Figure 1 title. Figure 1. Comparison of Functional Movement Screen score of Traditional Strength Training (TST) group and Functional Strength Training (FST) group
- Please revise the Figure 2 title. Figure 2. Comparison of Balance Error Scoring System (BESS) score of Traditional Strength Training (TST) group and Functional Strength Training (FST) group
- Please revise from FS to FST. Please revise from TS to TST.
- Errors in several word are observed. Please revise it after reviewing it all.
Author Response
Dear Reviewer
First of all, thank you for your comments and suggestions that allowed us to greatly improve the quality of the manuscript. We agree with all your comment. and we corrected point by point the manuscript accordingly.
- Please revise the Figure 1 title. Figure 1. Comparison of Functional Movement Screen score of Traditional Strength Training (TST) group and Functional Strength Training (FST) group
Figure 1. and figure 2. have been modified to increase the clarity
- Please revise from FS to FST. Please revise from TS to TST.
Abbreviation of functional strength training is revised from FS to FST and traditional strength training a revised from TS to TST
- Please describe information about the equipment used in this research. For example, you will need to include information such as model name, manufacturer, and country of manufacture. Like the information you put into the FMS tool.
Information about the equipment in this research was described as you specified.
- Errors in several word are observed. Please revise it after reviewing it all.
Sincere apologies for the poor English, we have corrected the grammatical and word mistakes
- Do you have any photos of functional strength training? If so, please include it.
The example of functional strength exercises and traditional strength exercises pictures were uploaded (Figure 1 and Figure 2)
- It would have been better if you included a control group in your study design. This part is a bit disappointing. Why didn't you include it in your research design?
As you comment, although it would be much more meaningful for results and discussion, we thought the number of participants is not adequately divided into three groups (TST, FS, and Control) because of gender differences.
Minor revisions are corrected carefully
Please see the attachment.
Sincerely,
Reviewer 2 Report
Dear Authors,
I have reviewed the following manuscript “Effects of Functional Strength Training on Functional Movement and Balance in Middle-Aged Adults”. Here are the comments:
Functional strength training (FST) has previously been well documented in terms of improving body strength, balance, and coordination in pre and post-pubertal age.
Major Comments#
Abstract requires re-writing. The methods section (in the abstract) should be written more.
Table II should be re-formatted. As the heading part of the table is on another page. Some p values are shown using a comma and some are shown using decimal. Use the decimal format and be consistent with it.
The results are shown in the table form. It will be more readable if the results are shown in the graphs and showing the significance using p values or asterisks in the graphs.
The discussion section could be improved by discussing more the previous studies on FST and how this current study adds new knowledge to the old concept of FST.
The English grammar and language of the manuscript need careful and complete proofreading.
Minor Comments#
Line 49: you do not need to write the FST abbreviation as you have already written it in the earlier section.
There should be a space between the sentence and the references. Please correct it everywhere in the manuscript. The manuscript has inconsistency in citing the reference style. Please be consistent.
Numbers used comma than decimals. Please change it to decimals.
Author Response
Dear Reviewer,
First of all, thank you for your comments and suggestions that allowed us to greatly improve the quality of the manuscript. We agree with all your comment. and we corrected point by point the manuscript accordingly.
We have changed all comma to decimal in table II.
To make results be more readable, they are shown in graph and showing the significance using an asterisk in graphs.
We made efforts to improve the discussion section
Sincere apologies for the poor English, we have corrected the grammatical mistakes and have asked an English native speaker to proofread entirely manuscript
Sincerely,

Reviewer 3 Report
In this manuscript, the authors investigate the effects of functional strength training on functional movement and balance in middle-aged adults. This is a very important area of research since poor balance can be seen in adults as young as 50 years of age.
In the abstract please change "...method for athletes, middle ages, elderly, to improve..." to "...method for athletes, middle-aged and older adults..."
Line 31: Replace "s" with "years" after 60.
Line 32: Replace "...decreased vary between 1-1,4..." with "...decrease varies between 1-1.4..."
Line 35: Replace "the middle ages" with "middle-aged adults" throughout the manuscript.
Lines 34-36. The whole sentence is poorly worded. Please rewrite.
Line 37: Delete the word "study."
Line 38: Replace "ageing's effect" with "effect of aging" throughout the manuscript.
Lines 39-41: Rewrite this sentence as "Strength training results in a reduction of adipose tissue and an increase in muscle mass and bone density."
Line 45: Again, "middle aged adults" rather than "middle ages."
Line 49: Change "...move multiple planes..." to "...move in multiple planes..."
Line 55: Change "...more effectively and target-oriented..." to "...in a more effective and target-oriented manner..."
Line 63: Change "...the middle-aged group..." to "...among middle-aged adults..."
Please provide details of the exercise experience of the volunteers prior to beginning the study. Were the volunteers physically active? or were they sedentary prior to beginning the study protocol?
The control group did no exercise in the 8 weeks before the study. Did they continue to perform no exercise during the study period? What was the physical activity level of the intervention group for the 8 weeks preceding the study? Were they also sedentary? This section is very unclear. I suspect the control group were instructed to perform no exercise during the 8 week period while the intervention group attended FST classes 3 times weekly. This must be explained more clearly.
Lines 86-88: These two sentences should be condensed into one.
Line 122: Change "p<0,05" to "p<0.05."
Line 154: "These results consist of previous studies" Do you mean that these results are consistent with previous studies?
Lines 156-157: No reference for the meta-analysis.
Line 157: References 40 and 41 are there but there is no sentence.
Line 159: Change "12,65 to 15,11" to "12.65 to 15.11."
Lines 161-165: Please remove any comparisons with standard strength training. You did not address this in your study.
Line 166: Delete "the."
Line 167: Change "...change observed in body mass..." to "...change was observed in..."
Line 168: Change "...strength training effects in the reduction..." to "...strength training leads to a reduction in adipose tissue, an increase in muscle mass..."
Lines 171-172: You did not address any comparisons between FST and regular strength training. This sentence has no reference and is speculation and should be deleted.
Lines 173-174. This sentence is incomplete and also needs a reference.
Line 175: Change "...strength effects in motor control..." to "...strength effect motor control..."
Line 194: Change "middle ages" to "middle-aged."
Any exercise intervention is likely to result in improved strength, increased muscle mass and bone density and loss of fat mass. A more relevant comparison would include a third group who performed a regular strength training regime so that you can compare FST with standard strength training. This was done in the studies by Yildiz et al (2019) and Pacheco et al. (2013) which you referenced.
It would have been interesting to compare males and females response to the training since you have the data.
Author Response
Dear reviewer,
First of all, thank you for your comments and suggestions that allowed us to greatly improve the quality of the manuscript. We agree with all your comment. and we corrected point by point the manuscript accordingly.
We explain the method section more descriptive way,
In the method section There were some explanation errors about the control and experimental groups that participated in the study due to our poor English. all statements were corrected and rewritten in an explanatory manner
Since our control group consists of individuals who do regular strength exercises, we did not change the comparison of traditional strength training and functional strength training in the discussion section.
Additionally, as you comment, we try to compare sex differences between participants although it would be much more meaningful for discussion the number of participants is not adequate for compare according to sex.
Minor revisions are corrected carefully
"Please see the attachment.
Sincerely

Round 2
Reviewer 2 Report
I am providing the following comments for the authors to address:
Cite figures 1 and 2 in the results section and explain the results presented in figures 1 and 2
Figure 1 and Figure 2 formats are not consistent. The size of Figure 1 and 2 should be the same
Figure 1: Put the top sentence in one line “Functional movement Screen” same as Figure 2. Numbers on graph bars should be aligned. At present, numbers are up and down
Complete figure legends are missing. Write the figure legends including what is represented in the figures, what is compared to what, and the p values represent significance between which groups.
Line 202: remove full stop before citations
Line 216: Put a space between exercise and (20)
Line 229-230: Please correct the grammar of the sentence
Line 237: Change the color of the full stop from blue to black
Put a space before writing any p values
Reference font style is not the same as the manuscript text
After the following corrections have been made, I recommend the editor(s) for considering its publication.
Author Response
Dear Reviewer,
Thank you for your comments and suggestions that allowed us to greatly improve the quality of the manuscript.
Complete figure legends are missing. Write the figure legends including what is represented in the figures, what is compared to what, and the p values represent significance between which groups.
Figures 1 and 2 are cited in the results section and explain
Figure 1 and Figure 2 formats are not consistent. The size of Figure 1 and 2 should be the same
Figure 1. and figure 2. have been modified and reorganized to increase clarity and consistency.
Complete figure legends are missing. Write the figure legends including what is represented in the figures, what is compared to what, and the p values represent significance between which groups.
Figure legends were written as you requested, and the p values were added to represent significance between which groups.
Please see the attachment.
Sincerely
Reviewer 3 Report
Dear Dr Guler,
thank you for addressing my concerns regarding this manuscript. I think it is much improved and easier to follow.
Author Response
Dear Reviewer
I grateful for your comments and suggestions that allowed us to greatly improve the quality of the manuscript
Sincerely,